# Prediction of nasal spray drug absorption influenced by mucociliary clearance

Yidan Shang[1,2], Kiao Inthavong[2]*, Dasheng Qiu[3]*, Narinder Singh[4,5], Fajiang He[1], Jiyuan Tu[2]

1 College of Air Transportation, Shanghai University of Engineering Science, Shanghai, China, 2 School of Engineering, RMIT University, Bundoora, VIC, Australia, 3 Department of Nuclear medicine (Positron Emission Tomography/Computed Tomography), Hubei Cancer Hospital, Wuhan, Hubei, China, 4 Department of Otolaryngology Head and Neck Surgery, Westmead Hospital, Sydney, Australia, 5 School of Medicine, University of Sydney, Sydney, Australia

* kiao.inthavong@rmit.edu.au (KI); hbpetct@163.com (DQ)

**Data Availability Statement:** All model files are available from the RMIT FigShare database (accession number(s) 10.25439/rmt.13550162.

**Funding:** This study was funded by the: 1. National Natural Science Foundation of China (Grant No.

## Abstract

Evaluation of nasal spray drug absorption has been challenging because deposited particles are consistently transported away by mucociliary clearance during diffusing through the mucus layer. This study developed a novel approach combining Computational Fluid Dynamics (CFD) techniques with a 1-D mucus diffusion model to better predict nasal spray drug absorption. This integrated CFD-diffusion approach comprised a preliminary simulation of nasal airflow, spray particle injection, followed by analysis of mucociliary clearance and drug solute diffusion through the mucus layer. The spray particle deposition distribution was validated experimentally and numerically, and the mucus velocity field was validated by comparing with previous studies. Total and regional drug absorption for solute radius in the range of $1 - 110nm$ were investigated. The total drug absorption contributed by the spray particle deposition was calculated. The absorption contribution from particles that deposited on the anterior region was found to increase significantly as the solute radius became larger (diffusion became slower). This was because the particles were consistently moved out of the anterior region, and the delayed absorption ensured more solute to be absorbed by the posterior regions covered with respiratory epithelium. Future improvements in the spray drug absorption model were discussed. The results of this study are aimed at working towards a CFD-based integrated model for evaluating nasal spray bioequivalence.

## Introduction

Nasal drug delivery has been recognised as a reliable alternative to parenteral routes since the richly vascularized nasal mucosa provides an effective route for drug absorption [1, 2]. Compared with traditional antibiotics and surgical interventions, nasal drug delivery offers safer treatment for localised nasal conditions such as rhinosinusitis, and delivery to the systemic circulation [3–5] and central nervous system [6–8]. However, limited knowledge of regional nasal deposition patterns [9, 10] and variations in the physico-chemical properties of the formulation make it difficult to predict bioequivalence of nasal spray drugs, which poses a challenge for regulators to assess the safety and efficacy of the products [3, 11].

81800096 to Yidan Shang, http://www.nsfc.gov.cn/english/site_1/index.html); 2. National Natural Science Foundation of China (Grant No. 91643102 to Jiyuan Tu, http://www.nsfc.gov.cn/english/site_1/index.html); 3. Garnett Passe and Rodney Williams Memorial Foundation Conjoint Grant 2019 (Inthavong-Singh granted to Kiao Inthavong and Narinder Singh, https://gprwmf.org.au/); 4. Australian Research Council (Project ID: DP160101953 granted to Jiyuan Tu, https://www.arc.gov.au/). The funders had no role in study design, data collection and analysis, decision to publish, or preparation of the manuscript.

**Competing interests:** The authors have declared that no competing interests exist.

Computational Fluid Dynamics (CFD) has been used to investigate deposition patterns of particles from spray devices [12–19]. These studies found low deposition efficiencies in the middle to posterior nasal cavity region and sinuses, where absorption occurs. Furthermore, deposition patterns influenced by spray nozzle insertion depth and orientations have been comprehensively investigated [16, 18]. Most commercial nasal sprays are formulated as suspensions with drug solutes suspended in a carrier liquid particle, and it takes time to diffuse through the mucus before being absorbed by the epithelium [20].

Rygg and Longest [21] proposed a nasal-mucus model that generated a mapped computational domain for mucus flow simulation. The model included mucociliary clearance, drug dissolution and convection in the mucus layer, making it possible to estimate drug absorption by epithelium. Rygg et al. [22] and Chen et al. [23] later linked a nasal-diffusion-absorption-convection (nasal-DAC) model with pharmacokinetic profiles to estimate nasal spray drug absorption and bioequivalence in clinical applications. The nasal-DAC model was applied to a nasal cavity without sinuses, and its surface walls were transferred to a 2D domain by calculating transverse perimeters.

Recently, the authors proposed a surface-mapping technique [24] and using this as a basis developed a 3D-shell model [25] to simulate the mucus velocity field on the nasal cavity wall. The 3D-shell model directly modelled the mucus flow on the 3D nasal cavity wall, which avoided geometrical deformation. The mucus velocity results along the wall were then mapped onto a 2D-domain for visualisation and comparison.

This study combined drug particle deposition with a mucociliary clearance model to evaluate its efficacy. The framework for this study is given in Fig 1 which integrates i) nasal airflow, ii) particle trajectories and deposition representing a nasal spray drug delivery from a release location, and iii) post-deposition motion driven by mucociliary clearance and drug solute diffusion through mucus. Drug absorption through the mucus during mucociliary clearance

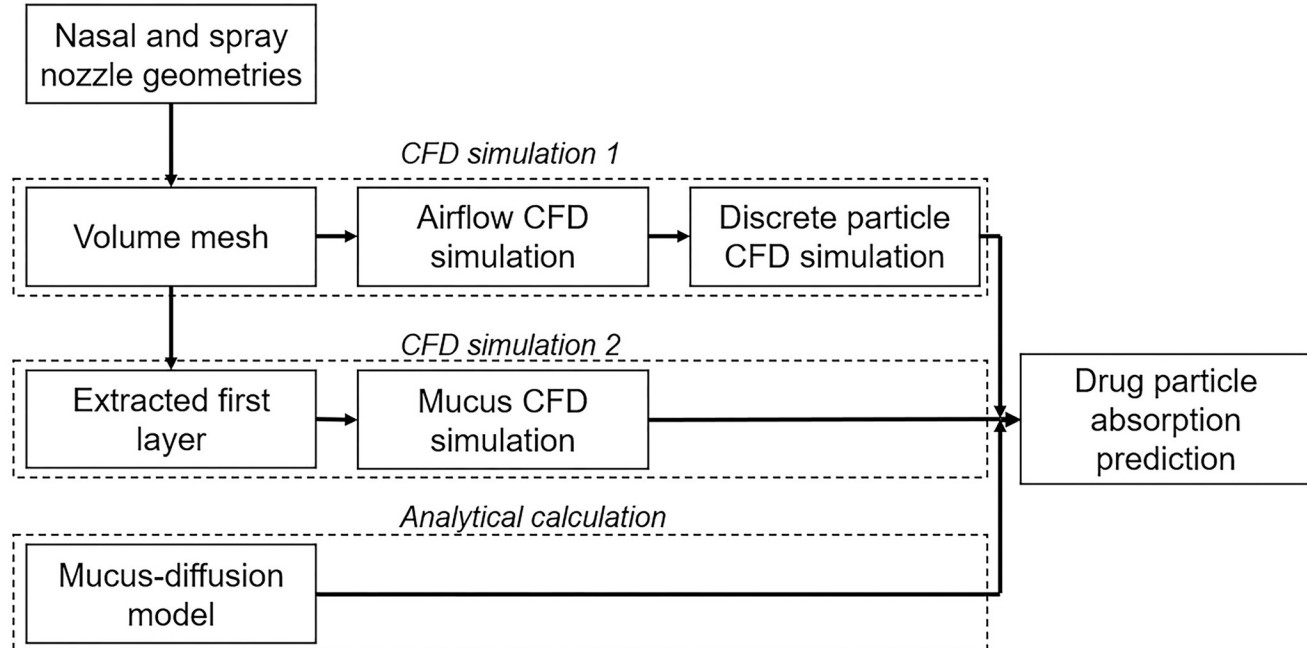

**Fig 1. Integrated modelling framework for drug deposition and diffusion.** The framework begins with CFD simulation 1—which computes the airflow field and spray particle deposition. CFD simulation 2 is a separate study that creates a mucus flow field; and the analytical calculation that combines data from the first two CFD simulations.

was predicted by applying a diffusion equation proposed in Erickson et al. [26]. This study advances current mucociliary clearance modelling by applying realistic spray droplets to determine preliminary deposition sites that allow more precise locations for evaluating drug absorption through nasal cavity wall. The mucus-diffusion modelling framework will provide a more realistic picture on therapeutic transmission to the epithelial cells.

## Results

### Nasal geometry and nasal spray droplet size distribution

The computational model of a human nasal cavity is shown in Fig 2, which was reconstructed from CT-scanned images of a 48-year-old Asian male, which included the external facial features [27–29]. This research was approved by the institutional review board at the Hubei Cancer Hospital (Hubei Province, China), and informed consent was obtained from patients. The model was separated into the left and right cavity, and nasopharynx. Each cavity was further divided into major anatomical regions: vestibule, main passage, septum, olfactory, maxillary sinus (Fig 2). Past studies have usually ignored the sinuses because the percentage of airflow entering them were negligibly small [9]. However, as it is a major source of mucus production, the sinuses should be included as part of the mucociliary clearance simulation. In this study, the maxillary sinuses and the lower ostiomeatal complex were included. Table 1 provides the surface areas of each anatomical region. The main passage occupied the largest percentage of surface area as it contained regions of the inferior/middle/superior meatuses and inferior/middle turbinates.

The droplet size distribution from the nasal spray actuation is given in Fig 3 which is defined by a Rosin-Rammler distribution with optimal mean diameter $78\mu m$, and spread parameter 2.7. The minimum and maximum diameters were $1\mu m$ and $150\mu m$, respectively, and number of diameters was set to 150.

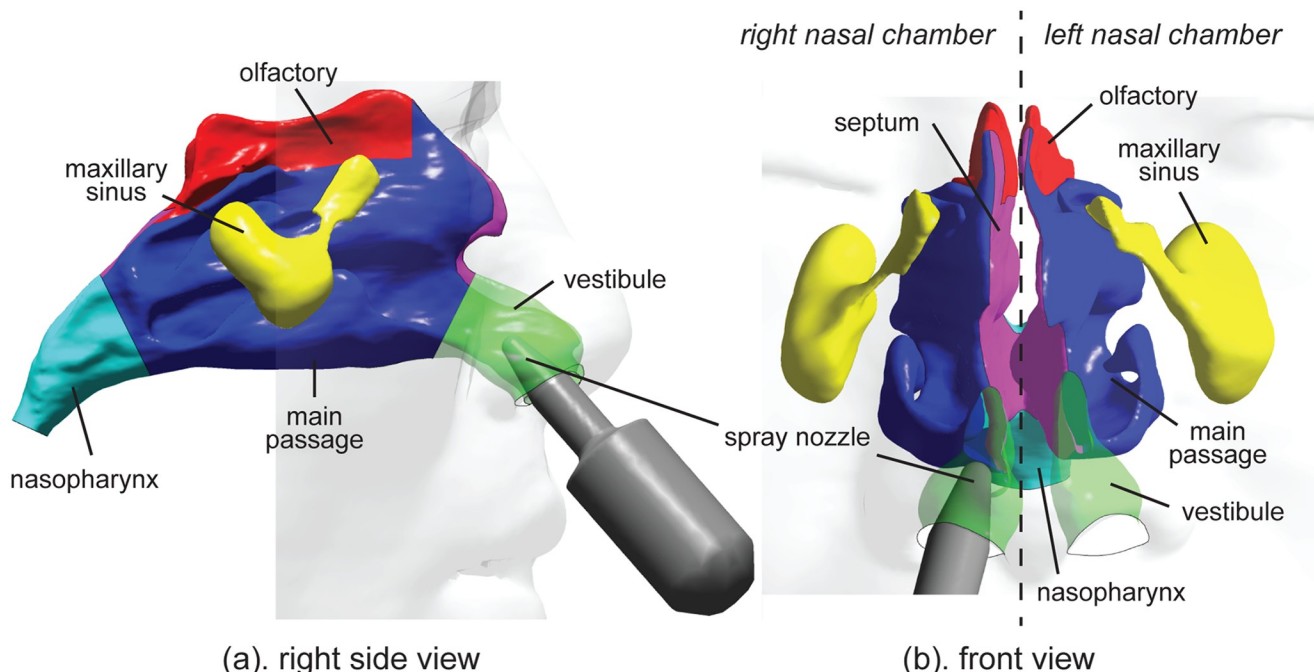

**Fig 2. Geometries of the spray bottle, the nasal cavity and its major anatomical regions including vestibules, main passage, septum regions, olfactory regions, maxillary sinuses and nasopharynx.**

**Table 1. Geometrical information of major anatomical regions.**

| Name of regions | Area ($cm^2$) | | Percentage (%) |
|---|---|---|---|
| | **Left chamber** | **Right chamber** | |
| Vestibule | 8.04 | 8.42 | 7.31 |
| Main passage | 56.3 | 54.1 | 49.1 |
| Septum | 20.8 | 20.4 | 18.3 |
| Olfactory | 9.12 | 9.87 | 8.44 |
| Maxillary sinus | 15.4 | 10.9 | 11.7 |
| Nasopharynx | 11.7 | | 5.20 |
| Total | 225.05 | | 100 |

### Airflow in the nasal cavity

The airflow characteristics were visualised using streamlines in the right nasal chamber, where the inhalation occurred through the gap between the nozzle and right nostril(Fig 4). Fig 4a showed that a jet airflow was formed beneath the nozzle. The jet was further accelerated until reaching its peak velocity (nearly 13 m/s) at the nasal valve, where the cross-sectional area reaches its minimum, and merged with other airflow streams that were traced back to the rest part of the right nostril.

In Fig 4b, where the septum was removed to properly reveal the airflow from medial view, illustrated that the main stream rapidly split into two streams as it passed through the nasal valve. The Stream 1 maintained its direction, flowed towards the superior passage and was further split into two sub-streams by the middle turbinate: stream 1-1 flowed medially, bounced forward after impacting the superior turbinate, forming a low-velocity recirculation and merged with stream 1-2, which flowed through middle meatus. The stream 2 flowed through the inferior meatus and then merged with stream 1 at the choanae, before exiting nasal cavity through the nasopharynx.

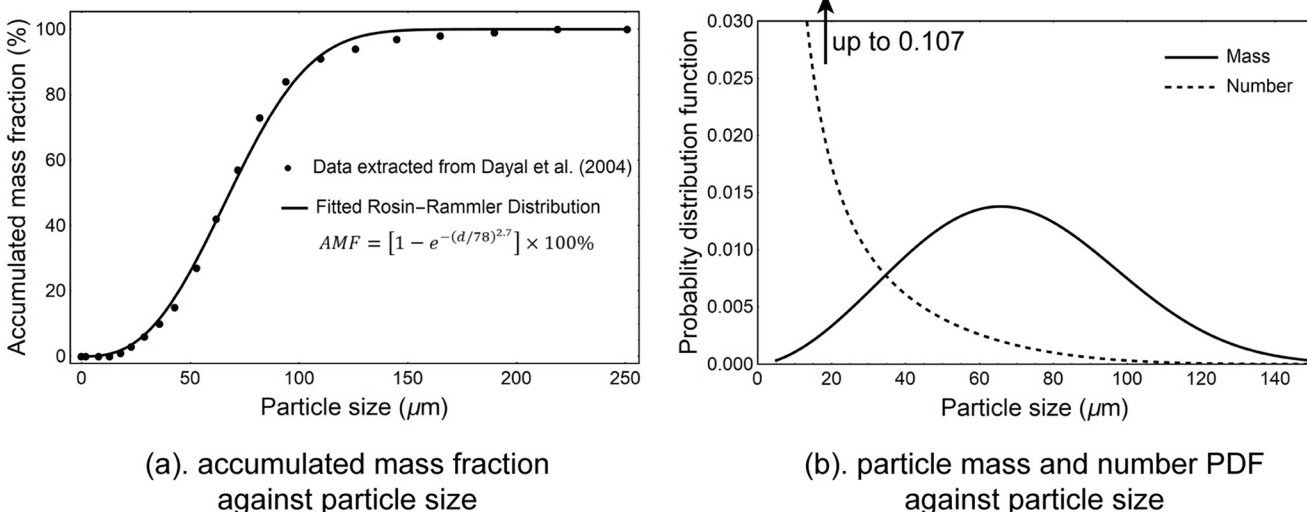

(a). accumulated mass fraction against particle size

(b). particle mass and number PDF against particle size

**Fig 3. Spray drug particle size distribution used in the CFD simulation.** (a) Measured data from Dayal et al. [30] showing cumulative mass fraction, fitted using Rosin-Rammler Distribution. (b). The corresponding mass and number probability distribution function over particle size. The range of particle size released was $1 - 150 \mu m$.

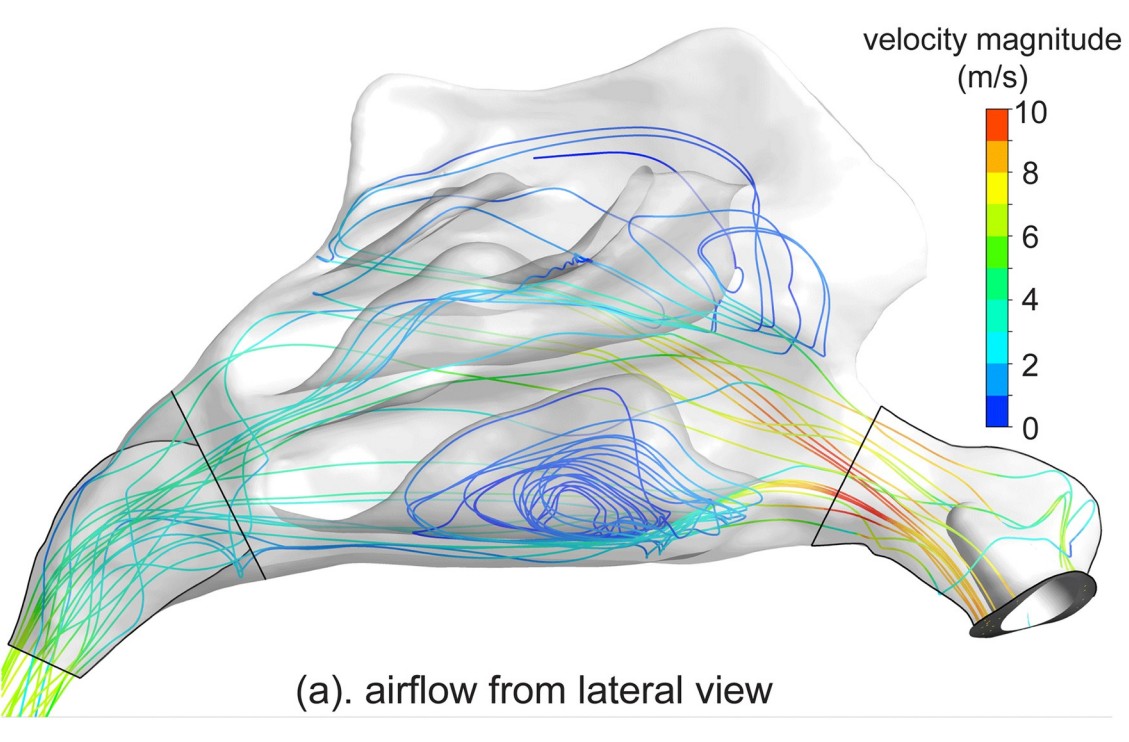

(a). airflow from lateral view

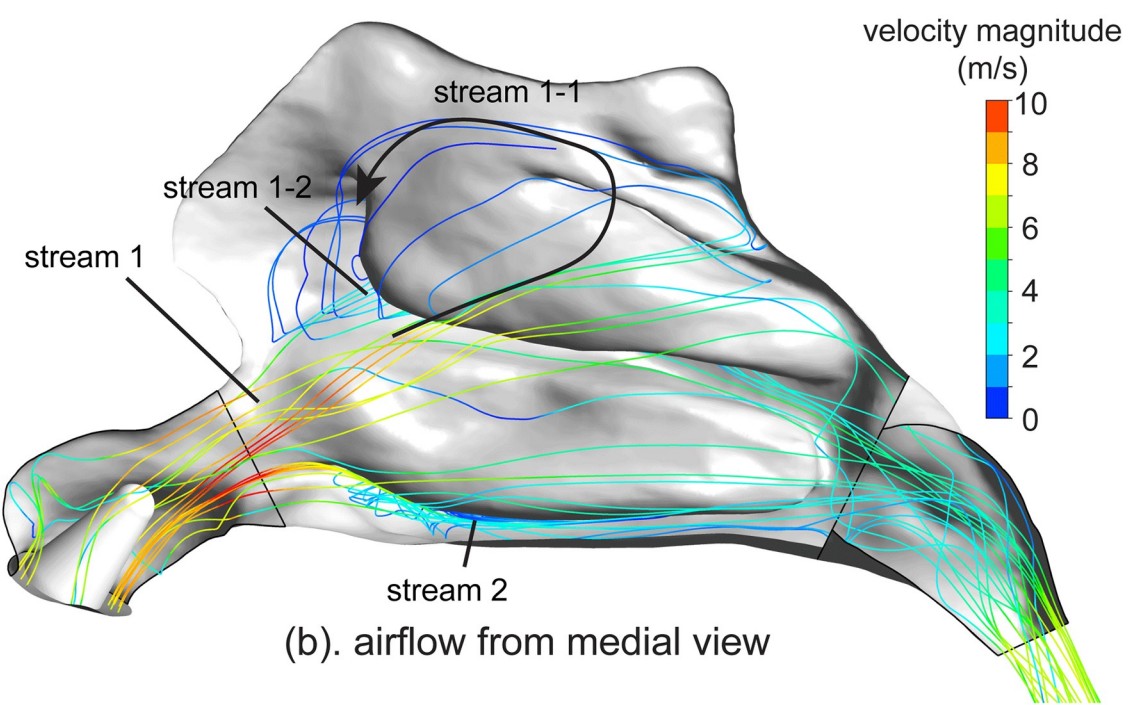

(b). airflow from medial view

**Fig 4. Airflow streamlines passing through the right chamber of the human nasal cavity.** Streamlines are coloured with velocity magnitude. a). lateral view with transparent nasal walls. b). medial view with septum removed.

## Particle deposition patterns

Particle deposition from a nasal spray device in the right chamber is shown in Fig 5. The 3D view (Fig 5a) indicated that most particles directly impacted on the anterior nasal cavity region

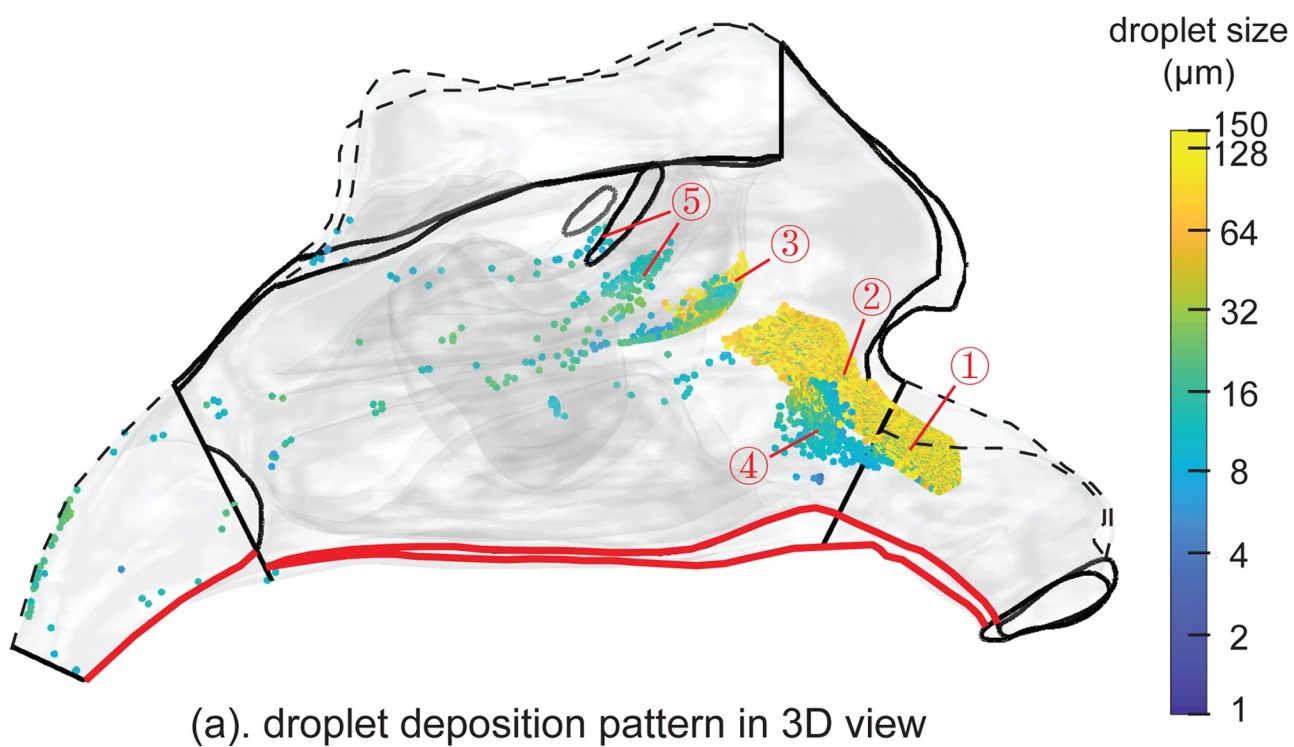

(a). droplet deposition pattern in 3D view

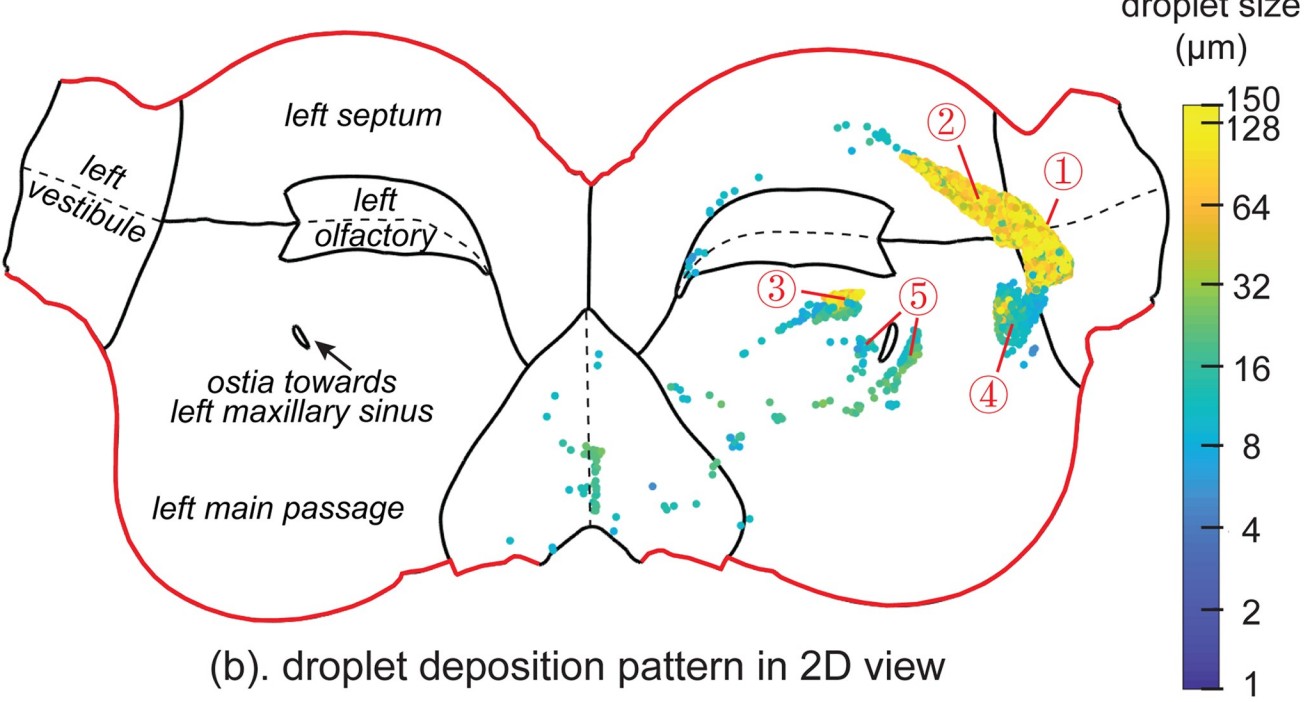

(b). droplet deposition pattern in 2D view

**Fig 5. Particle deposition patterns coloured by particle size ranging from 1μm to 150μm.** (a). Particle deposition distribution in 3D view of the right chamber. Nasal walls were set transparent to visualise particle deposition locations. (b). Particle distribution in the unwrapped 2D view. All deposited particles were at the right cavity where the spray particles were released. There are three hot-spots of deposition caused by direct impaction. The first hot-spot mainly depicted by large droplets with size >30μm was a band-like region that across vestibule and septum (① and ②), the second one is located at the tip of the middle turbinate (③) and the third one mainly hosts small droplets with size <30μm was a more concentrated area that across vestibule and lateral cavity (④). Another deposition hot-spot for nearly 30μm particles is located around the maxillary ostia (⑤).

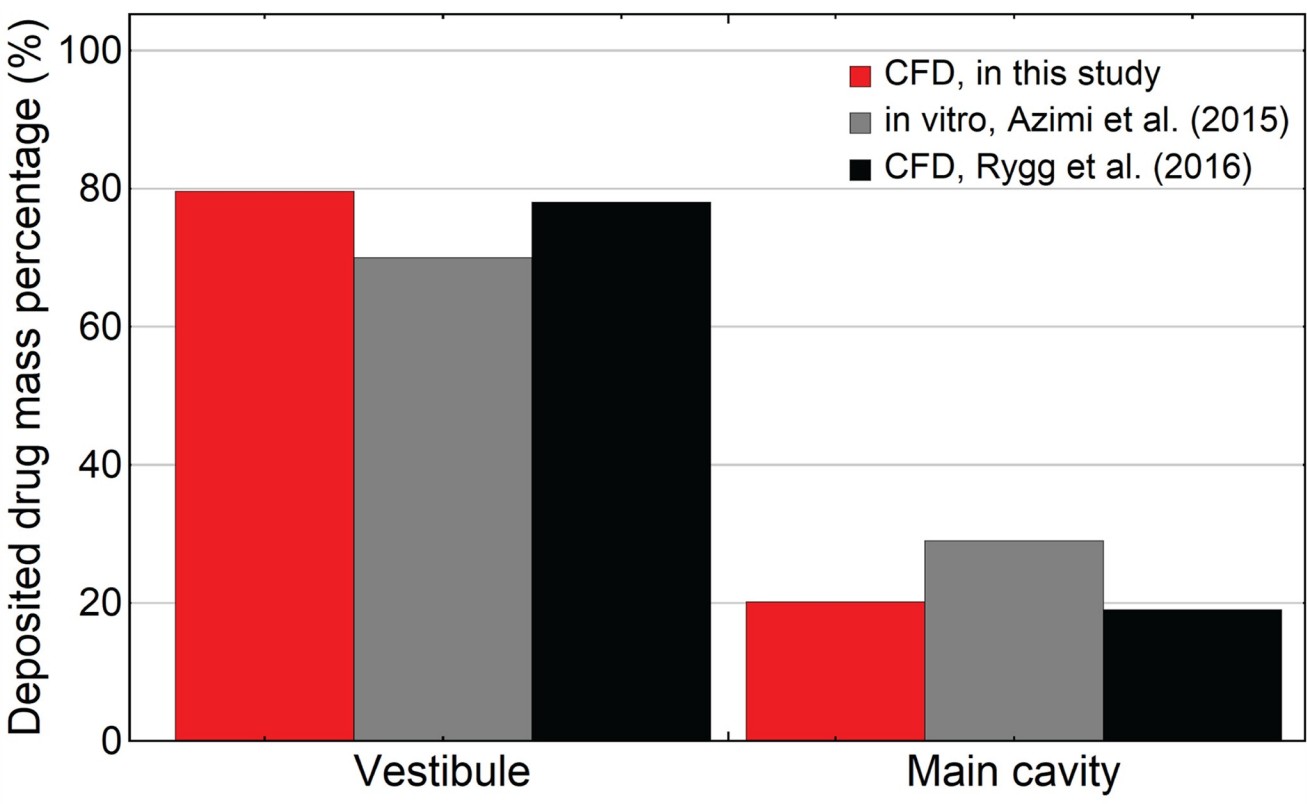

**Fig 6. Regional particle depositions in the vestibule and main cavity and comparisons with reported in-vitro measurements from Azimi et al. [31] and reported CFD simulation results from Rygg and Longest [21].**

in front of the particles' release location. Those regions include the posterior vestibule, anterior part of the main cavity and the anterior tip of the middle turbinate. Only smaller particles ($< 20\mu m$) penetrated beyond the anterior nasal cavity and deposited in the posterior regions such as nasopharynx.

The unwrapped 2D nasal cavity surface is shown in Fig 5b, with anatomical regions labelled in the left chamber labelled (mirroring the right chamber). The nasal cavity wall was cut along the shared boundary (labelled in red curves) of the nasal passage and the septum. The dashed curves represent ceilings of the vestibule, olfactory and nasopharynx regions, which can be identified in the 3D view (Fig 5a). There was no deposition in the maxillary sinuses, and therefore these regions were removed, but the ostia openings were retained. There are three deposition hot-spots located at the posterior vestibule (①), anterior septum (②) and the anterior tip of the middle turbinate (③) for large particles with sizes $>60\mu m$, whereas the anterior main nasal passage (④) are the main deposition locations for relatively smaller particles with sizes nearly $20\mu m$. There is another minor hot-spot observed near the maxillary ostia (⑤) for nearly $30\mu m$ particles.

The deposition efficiency on the vestibule wall and the main passage walls were compared with in-vitro measurements [31] and CFD results [21](Fig 6). Our results produced 100% deposition efficiency in the nasal cavity, with approximately 79.8%, 19% and 1.2% particle mass deposition in the vestibule region, main passages and nasopharynx, respectively. The deposition efficiency in the vestibule region is 2.3% higher than the results from Rygg and Longest [21] and 14% higher than from Azimi et al. [31].

## Mucus velocity distribution

Fig 7a shows the mucus velocity field from the nasal vestibule to the nasopharynx in 3D view, and in Fig 7b on the surface-unwrapped nasal wall. The mucus velocity magnitude is from 1.2 mm/min to 30 mm/min that represents the range of 0.2 and 5 times of the average mucus velocity magnitude of 6 mm/min over the whole nasal cavity wall (excluding vestibules and sinuses). The colourmap was set to logarithm scale, and unit vectors were attached to indicate the mucus flow direction.

In the 3D view (Fig 7a), the nasal geometry was split into left and right chambers, and for each chamber, both lateral and septal sides were presented. Similar mucus velocity distributions were found between each chamber, despite some minor geometrical differences between them. Low mucus velocity regions were located at the vestibules, anterio-superior regions, the olfactory region, and the distal maxillary sinuses, while high mucus velocity regions were concentrated at the nasopharynx and the maxillary ostium that bridges the maxillary sinuses and middle meatuses. The velocity magnitude in the septum region was predominantly around the average velocity of 6 mm/min and was relatively similar when compared with the large velocity variations on the main passage walls.

The mucus velocity distribution in the unwrapped domain (Fig 7b) shows the mucus blanket in each chamber gradually accelerating from the anterior regions, then moving posteriorly, before merging at the choanae and further accelerate in the nasopharynx where the circumference shrinks rapidly. A discernible radial flow pattern originating from the maxillary ostia was observed over each nasal chamber. The anterior part of the radial flow initially moved forward, reaching the nasal valve, then turned back and split into two streams laterally and towards the septum. The two streams merged with the posterior part of the radial flow at the choanae and exited at the nasopharynx The mucus produced from the maxillary sinus significantly influenced the mucus flow on the main passage and created regions of recirculation on both the lateral and septal regions of the nasal chamber.

Fig 8 shows the deposited particles transported by the mucociliary clearance over 30 minutes. Particles were labelled in different colours representing their deposition locations. The particles on the nasopharynx region (cyan) were the first to be cleared within 1 minute, followed by particles on the olfactory region (yellow) which were cleared within 5 minutes. The particles located on the septum (pink) were cleared within 10 minutes, which were transported through the inferior nasal chamber. Particles depositing on the main passage (blue) were transported in three groups. The first and second group of particles, located at the superior-posterior region and posterior to the maxillary ostia, were cleared in 15 minutes and 10 minutes, respectively, as they moved directly towards the nasopharynx. However, particles located anterior to the maxillary ostia first moved towards the vestibule (away from the nasopharynx exit) and then turned back and moved along the floor of the nasal chamber, and eventually, most particles were cleared in 30 minutes. Particles deposited at vestibule (green) took the longest time to clear. They firstly split into two streams moving in opposite directions, namely laterally and towards the septum, then merged with the mainstream at the floor of the nasal chamber, indicated by the shared boundary of the septum and main passage (refer to red curves in Fig 5). After 30 minutes, the considerable amount of particles that deposited on the vestibule were not cleared by the mucociliary clearance.

## Drug solute absorption

Drug absorption efficacy by different anatomical regions can provide insight to clinicians for targeted drug delivery. The drug solute originally in a carrier particle diffuse through the mucus layer as they are transported by the mucociliary clearance. For a solute travelling

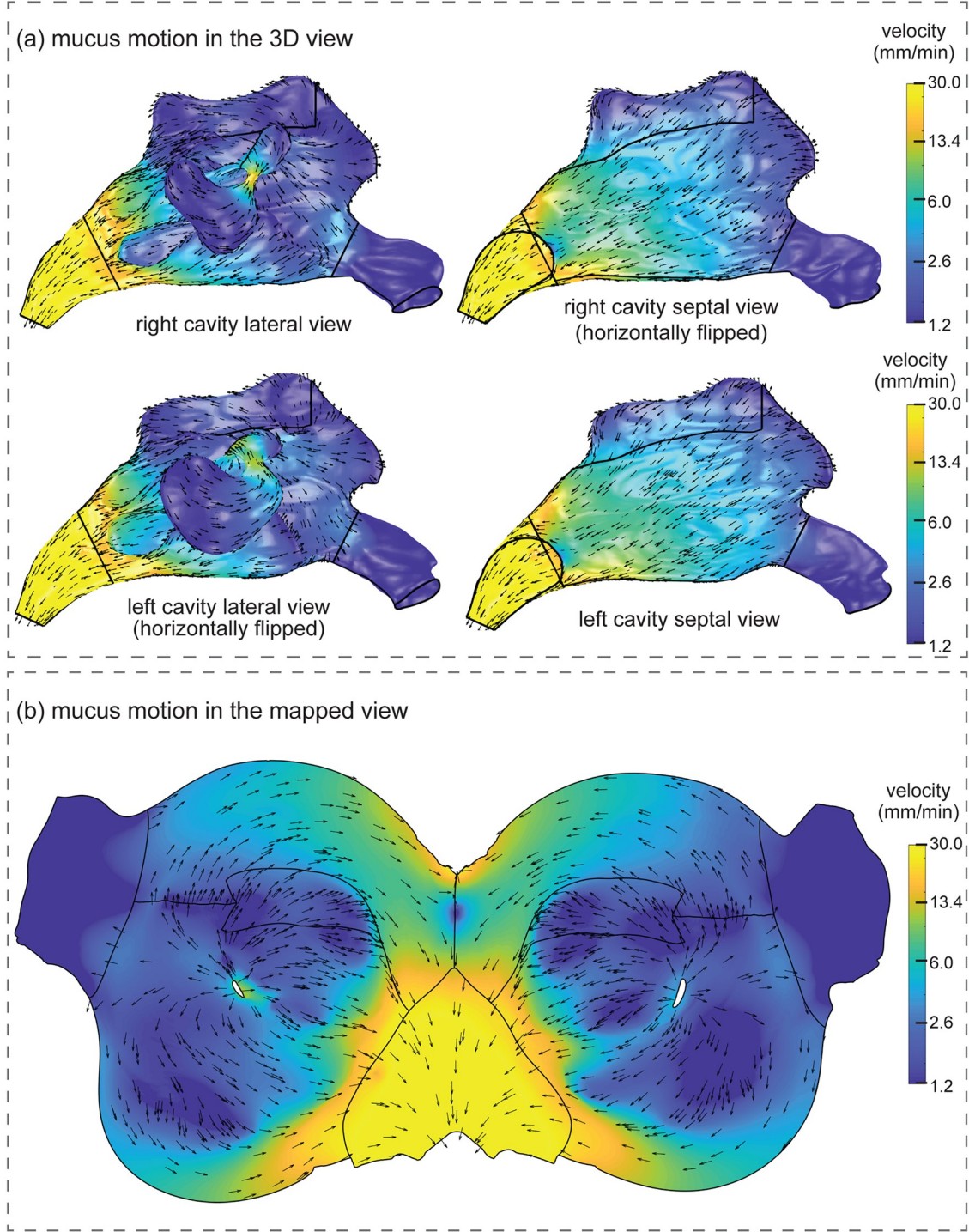

**Fig 7. Simulated mucus velocity distribution on the nasal cavity wall.** (a) Mucus velocity visualised on both lateral and septal sides of the left and right nasal cavities separately. (b) Mucus velocity visualised in a surface-unwrapped domain.

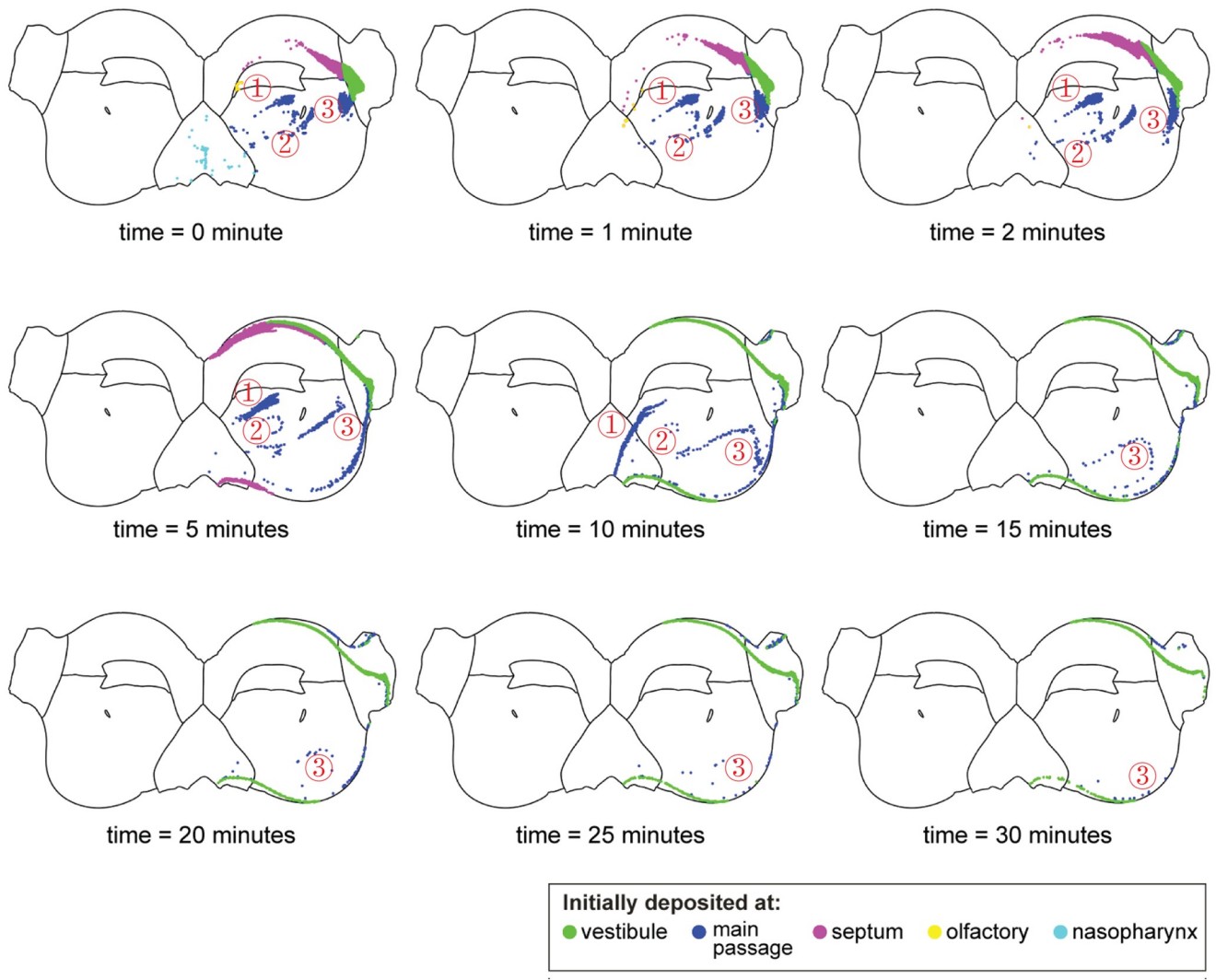

**Fig 8. Post-deposition particle transport over time.** The particles were coloured by deposition location.

through a particular anatomical region, the entry time $t_{in}$ and exit time $t_{out}$ were recorded to calculate its absorption $A_m$,

$$A_m = \alpha \times m \times [A(t_{out}) - A(t_{in})]　\qquad (1)$$

m is the carrier particle mass. $\alpha$ is the drug solute mass fraction that is assumed a constant. The regional drug solute percentage $A_r$ is calculated by summing all drug solute mass absorption and then normalised by total drug solute mass,

$$A_r = \frac{\sum_{i=1}^{n_r} m_i \times [A(t_{i\_out} - A(t_{i\_in}))]}{\sum_{i=1}^{n} m_i} \times 100\% \qquad (2)$$

$n_r$ is the number of particles travelling through a particular region and n is the total number of particles depositing on the nasal walls.

Fig 9 represents the total and regional absorptions as a function of drug solute radius. The total deposition steadied around 20% for solute radius smaller than $60 nm$. This gradually

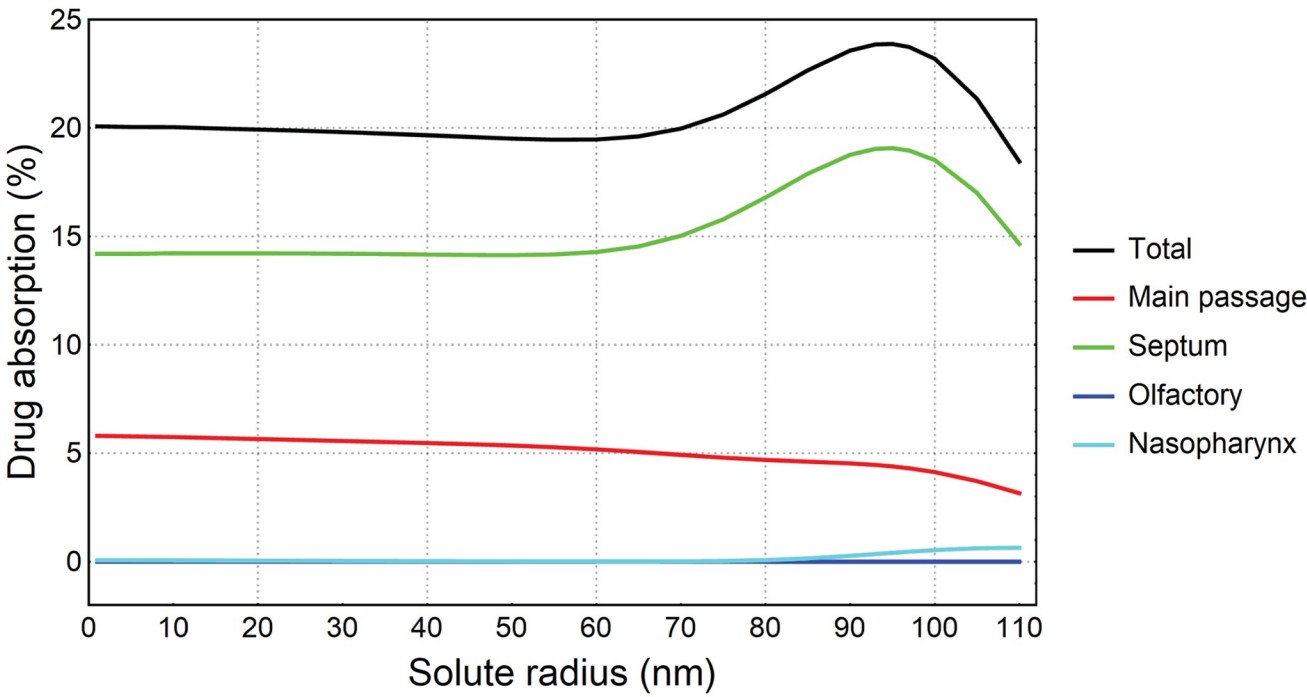

**Fig 9. Total and regional drug absorption over drug solute radius.**

increased as solute radius became larger and peaked at 24% when solute radius reached 95$nm$. For a solute radius larger than 95$nm$, the total absorption dropped and reached 18.5% at solute radius 110$nm$. The absorption curve of the septum region was similar to the total absorption and consistently absorbed the largest percentage of drug solute (14.2%–19.1%). The second-largest absorption region was the main passage. It showed a decreasing trend as the solute radius increased, with absorption of 5.8% for 1$nm$ solute and 3.2% for 110$nm$ solute. The drug solute absorbed by olfactory and nasopharynx regions were negligible, with the highest absorption 0.64% for 110$nm$ solute absorbed by nasopharynx region.

Fig 10 illustrates the total drug absorption produced from the initial locations/regions of the deposited particles. Although the vestibule region does not absorb drug solute, drug particles deposited on the vestibule were transported to other regions that had respiratory epithelium where drug absorption occurs. For example, particles depositing on the vestibule were transported to other regions where absorption took place. For solute radius smaller than 50$nm$, initial particle deposition on the septum and main passage produced the highest (nearly 14%) and second-highest (nearly 6%) contributions to the total absorption (nearly 20%), whereas the contributions from initial particle deposition on the vestibule, olfactory and nasopharynx were negligible. As the solute radius increased to 80$nm$, the contributions from the septum and main passage slightly dropped but the contribution from vestibule soared exponentially. For solute radius larger than 80$nm$, the contribution from the vestibule continued to increase until it reached 12% when solute radius was 105$nm$, exceeding contributions from main passage and septum at solute radius 85$nm$ and 95$nm$, respectively.

## Discussion

In the authors' previous study [25], the maxillary sinuses were excluded from the nasal cavity model and the mucus production was replaced with a representative mucus injection rate at

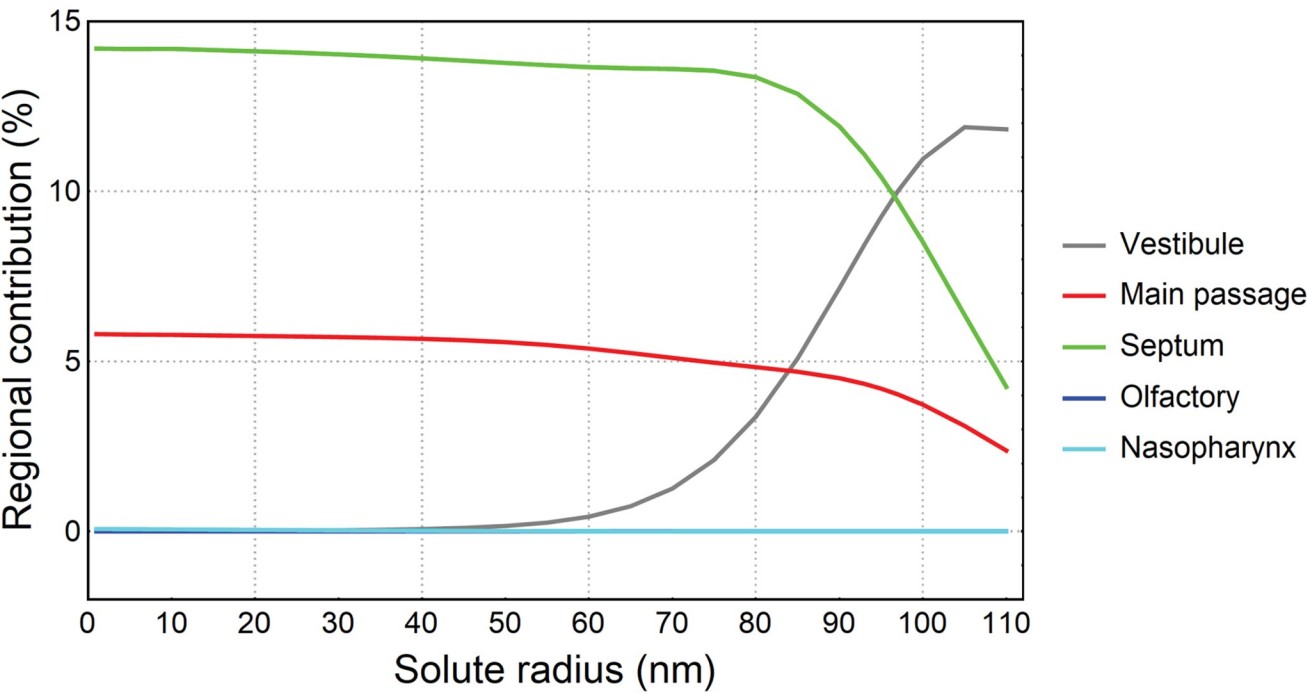

**Fig 10. Total drug absorption produced from initial deposition regions, for different solute radius.**

ostium. The mucus velocity field simulated in this study is consistent with the 3D-shell model in the previous work [25] except for a slightly stronger radial mucus flow observed around ostium. Therefore the mucus velocity distribution can be considered validated. In this study, the vestibule regions were included in the nasal geometry which was not present in Shang et al. [25]. This provided a more realistic mucus flow where an area of mucus velocity formed near the nasal valves, agreeing with clinical observations.

The clearance-diffusion model provided an approach to evaluate total and regional spray drug absorption for drug solute with different effective macromolecular radius. The Fig 11a illustrates a decreasing trend of diffusion coefficient against the solute particle radius, and information of some typical macromolecular were plotted for comparison. As a result, The drug absorption rate is dependent on the effective radius of the drug solute. Fig 11b shows that for 10$nm$ solute, the absorption reaches 100% in less than 10 seconds. As the solute radius increases, the time for solutes to diffuse is longer. For 90$nm$ solute, almost no absorption occurs in the first 20 seconds, followed by a gradual absorption increase, which reaches 97% at 500 seconds after deposition. The diffusion equations and the Obstruction-Scaling model determined that it took longer time to absorb drug solutes with larger radii, thus may lead to lower drug absorption as the deposited drug particles are transported by the moving mucus out of the nasal cavity region during the diffusion process. On the other hand, slow diffusion rates caused by large solute radius enables more particles that deposited on the vestibule to be absorbed by respiratory epithelium rather than being lost in the vestibule region. This mechanism is evident in Fig 10 where the drug absorption contributed by particles deposited on the vestibule increases exponentially when the solute radius is larger than 50$nm$. These two conflicting trends lead to a peak of drug absorption at solute radius 95$nm$. There exists a gap between the absorption fraction and the real dosage. Eq (3) is required to link the drug

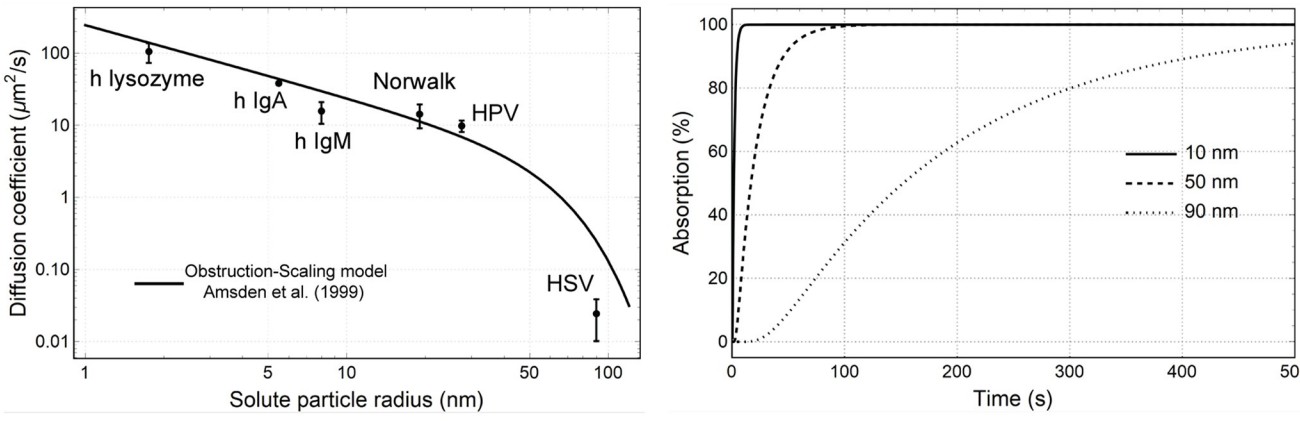

**(a). Diffusion coefficient in mucus gel over solute radius**

**(b). 10 nm, 50 nm and 90 nm solute particle absorption over time**

**Fig 11. Properties for dissolved drug particle diffusion through mucus gel layer.** (a) Solutes' diffusion coefficient were calculated using Obstruction-Scaling model and plotted with typical macromolecular measured in the study of Olmsted et al. [32]. h represents human proteins, (b) Absorption functions over time predicted by Eq (11) were plotted for solute particles with an effective radius of 10$nm$, 50$nm$ and 90$nm$.

absorption fraction with drug dosage in the clinical applications.

$$dosage = \frac{4}{3}\pi \times c \times \sum_{i=1}^{n}\left[A_i(+\infty) \times r_i^3\right] \tag{3}$$

where $c$ is the solute concentration, $A_i(+\infty)$ is the final absorption fraction of the $i$th deposited spray droplet and $r_i$ is its radius.

The drug absorption process occurs inside the mucus layer which was modelled in a separate domain to the nasal airflow (see Fig 1). It is expected that breathing would influence the mucus layer surface velocity, but not the absorption process directly. However, since the breathing is cyclic with inhalation and exhalation, we expect the mucus surface velocities inducing increased forward and backward mucus motion would be negated over each cycle and therefore its influence on the absorption process is negligible over time.

## Materials and methods

### Mesh generation and boundary conditions

The CFD domain of the nasal cavity was created with polyhedral mesh elements, refined near the boundaries, and in regions with high curvature and thin gaps, using Ansys-Fluent-Meshing v19.3 (Fig 12). Five prism layers were created on computational boundaries. One prism layer that was attached on the nasal cavity walls was extracted to form the computational domain of the mucus layer, which covered walls of the vestibule, the main cavity and the outlet extension pipe. A mesh independence test was performed for meshes under three refinements (e.g. coarse, medium and fine) and the optimal mesh (medium) with 2.02 million cells was selected for computation. The maximum skewness of the selected mesh was 0.70.

A spray device model was created based on a typical commercial nasal spray device used in other studies [16, 33, 34], which was inserted into the right nostril with a depth of 10 mm, suggested by Azimi et al. [31]. The spray bottle was inserted into the right nasal chamber at an insertion angle of 30° from horizontal (forward tilt), a spray cone angle of 50°, and an initial spray velocity of 10 m/s.

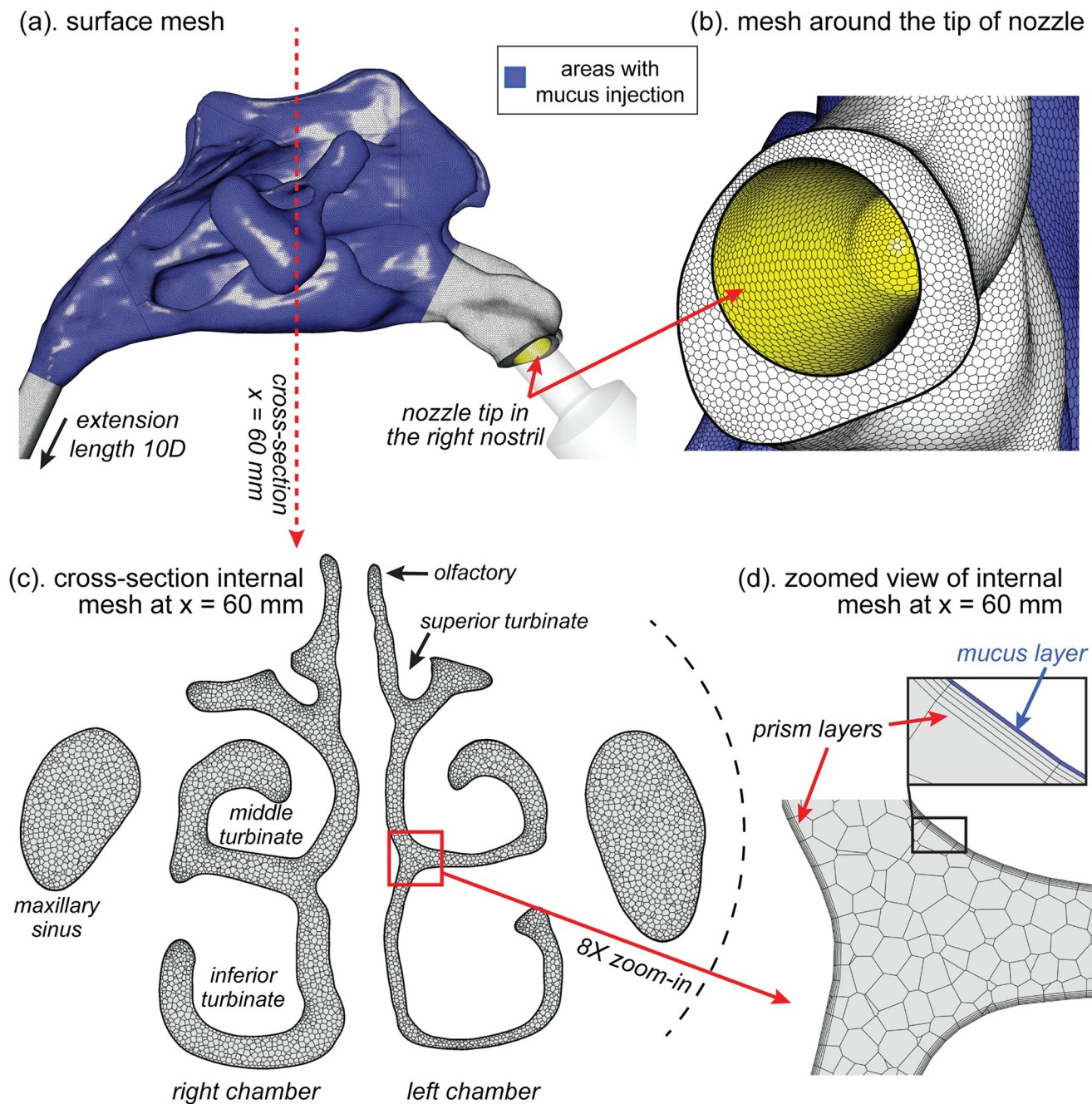

**Fig 12. The CFD nasal cavity model contained polyhedral mesh elements.** (a) Surface mesh on the vestibule, the nasal cavity and the outlet extension. All surface areas were covered with a 10 μm thick mucus layer, but the source term for mucus injection was applied in the main nasal cavity only (coloured in blue). An extension from the nasopharynx exit was extruded with a length of 10 times its diameter to ensure fully developed flow at the outlet. (b) Refined surface mesh around the nozzle tip. (c) Cross section slice located at 60 mm away from the tip of nose showing the internal mesh. It also shows major anatomical regions including superior/middle/inferior nasal passages, superior/middle/inferior turbinates, olfactory regions and maxillary sinuses. (d) 8X zoomed view of the internal mesh containing polyhedral cells, prism layers and thin mucus layer (coloured in blue).

An inhalation flow rate of 20.2 L/min was used which corresponded to a peak inspiratory flow rate during a slow cyclic inhalation condition. The outlet boundary condition was set to a velocity drawing air from the nasal cavity. The right nostril was set as a pressure inlet (0 Pa), and the left nostril was blocked to represent a deliberate occlusion by a patient closing the outer vestibule shut.

The DPM (Discrete Particle Method) condition at the nasal walls was set to "trap", where the Lagrangian particle tracking was terminated the individual particle came into contact with the nasal walls. The rest of the boundaries were set to "escape". Two hundred particles streams per particle bin (150 particle bins in total) in the Rosin-Rammler distribution were released from a solid circle that was 2.5 mm from the nozzle tip so that there was 30,000 particle stream in total. The 6th-order Runge-Kutta scheme was used for particle tracking.

For the mucus computational domain, a source term was added to the continuity equation for the mucus regions:

$$\rho(\nabla \cdot \vec{v}) = S_{mucus} \tag{4}$$

The value $S_{mucus}$ was determined by allowing it to produce an average mucus velocity magnitude of 6 mm/min [35] on the main cavity wall that excluded vestibules and sinuses. No mucus source term was applied to the vestibules and the extension regions. The outlet at the extension-mucus domain was set to a pressure boundary condition (0 Pa) and all other boundaries of the mucus domain were a slip-wall with zero shear stress. The accumulation of the mucus source term and the incompressible assumption pushed the mucus backwards to the nasopharynx region. The low mucus velocity led to a Reynold's number as low as 1e-4, hence the mucus flow was considered as laminar flow. The mucus velocity distribution was calculated iteratively using second-order discretisation method and the SIMPLE scheme.

## Governing equations

The Reynolds Stress Model (RSM), which has the advantage of capturing turbulent anisotropy near walls in a complex flow, was selected to predict the transitional flow using the SIMPLE algorithm for pressure-velocity coupling. The accuracy of RSM model has been evidenced to cover low and high Reynold's number ranges. In this study, the RSM simulation was considered converged when all residuals reached 1e-5.

The released drug particles were tracked using the Lagrangian discrete phase model (DPM) where the equation of motion for each particle given,

$$\frac{du_i}{dt} = f_D + f_G \tag{5}$$

$u_i$ is particle's velocity at ith coordinate, $f_D$ and $f_G$ are drag force and gravity force per unit mass. The drag force is determined by Stokes drag law ([36]),

$$f_D = \frac{18\mu}{d_p^2 \rho_p} \frac{C_D Re_p}{24} (u_i^{air} - u_i) \tag{6}$$

where $\mu$ and $u_i^{air}$ are viscosity and velocity of air, $d_p$ and $\rho_p$ are size and density of a particle. $Re_p$ is the particle Reynolds number, and $C_D$ is the drag coefficient,

$$Re_p = \frac{\rho_p d_p |u^{air} - u|}{\mu} \tag{7}$$

$$C_D = a_1 + a_2/Re_p + a_3/Re_p^2 \qquad (8)$$

where $a_1$, $a_2$ and $a_3$ are empirical constants [37].

Diffusion coefficients as a function of solute macromolecular radius were calculated using the Obstruction-Scaling model, (plotted in Fig 11a). The results showed good agreement with some typical macromolecular diffusion coefficients that were measured by Olmsted et al. [32]. The diffusion coefficient steadily decreased as solute particle radius increased from $1nm$ to $50nm$. For drug solutes with a radius larger than $50nm$, the diffusion coefficient dropped significantly matching the experimental results of $59 - 1000nm$ polystyrene bundles of mucins [32].

## Nasal spray

Nasal drug delivery from a spray device was performed using inhalation and initial particle conditions from Dayal et al. [30], Azimi et al. [31] and Tong et al. [16]. The particle-size distribution was extracted from of Dayal et al. [30] where a Nasonex spray bottle at an actuation distance of 3 cm under a 4.5 kg force was used (Fig 3). The data was fitted by the Rosin-Rammler distribution with optimal mean diameter $78\mu m$, and spread parameter 2.7. The minimum and maximum diameters were $1\mu m$ and $150\mu m$, respectively, and number of diameters was set to 150.

## Mucus-diffusion model for dissolved particles

The nasal cavity is covered with a thin mucus layer that traps foreign particles and transports them towards the oropharynx. The mucus gel composition is primarily water (90–95%) and protein fibre networks called mucin (3%). The remainder (about 2%) includes electrolytes, serum proteins, immunoglobins and lipids [38]. Experimental studies indicated that the adhesive mucin network significantly increased the mucus viscosity up to 10,000 times of water [39]. The mucus blanket comprises two distinct layers (Fig 13) with a total thickness of 5 $- 15\mu m$ [35, 40] where the upper layer is a thick gel containing adhesive mucin networks, and the lower layer is the watery periciliary layer (about $5\mu m$ thick [40]) that coordinates rhythmic cilia beat.

Cilia located on the epithelium surface rhythmically propel the mucus blanket backwards. The mechanical coupling between mucus and cilia maintains a constant thickness since a thicker or thinner layer will lead to impaired mucociliary clearance function [40]. The mucus layer velocity field flowing over the nasal epithelium was set with a constant mucus density (1,000 kg/m3) and mucus viscosity (12 Pa.s), suggested in previous experimental [41] and numerical [21, 25] studies. For particle diffusion through the mucus, the upper gel layer was set to a constant thickness of 10 μm and the lower periciliary layer thickness was set at $5\mu m$. The high viscous gel eliminates convection in the upper gel layer, thus we ignored the velocity along the wall-normal direction, following Rygg and Longest [21]) and Shang et al. [25]. The cilia propelling motion leads to a complex flow in the periciliary layer, which is difficult to predict. Since the periciliary layer does not host the mucin network, the diffusion model was simplified with the assumption that the solute reaching the periciliary layer is immediately absorbed by the epithelium.

The absorption rate of deposited particles was calculated through the moving mucus velocity field. For simplicity, the particles were assumed to immediately dissolve into macromolecules after depositing on the upper surface of the mucus layer, and the drug solute within the carrier droplet particle was then released locally. The drug solute's diffusion through the

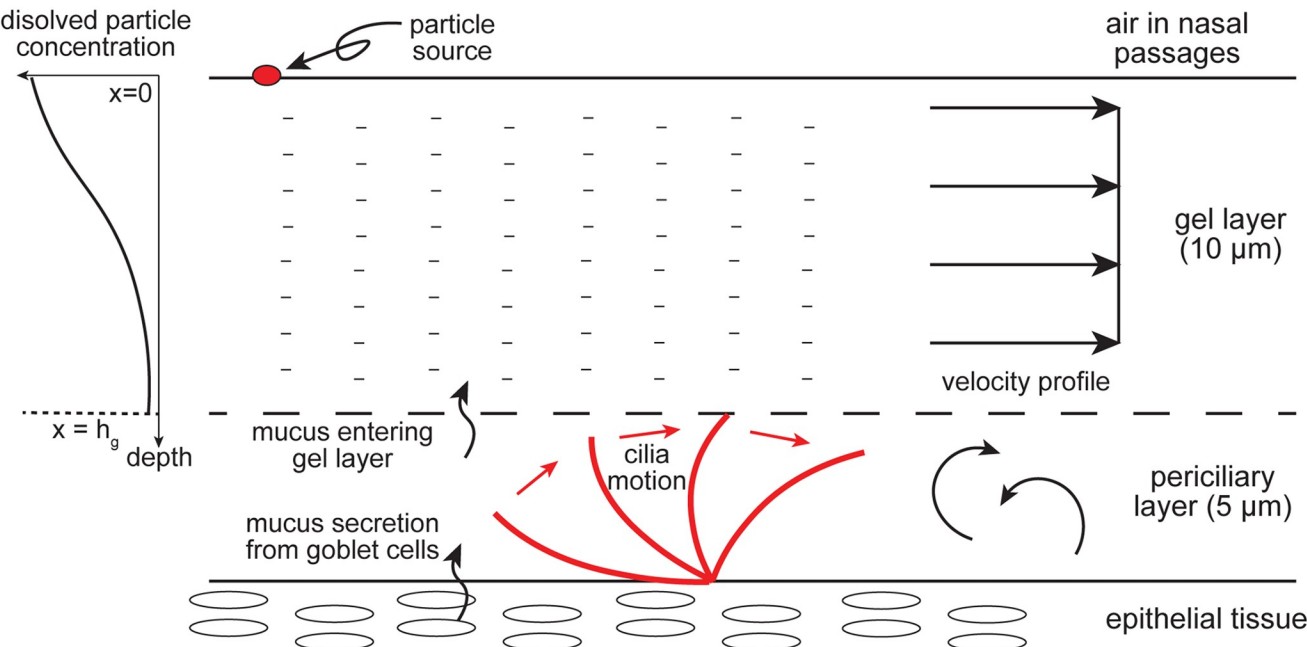

**Fig 13. Schematic of the mucus layer structure and dissolved drug particle diffusion through the gel layer.** The mucus model and the diffusion model were reproduced from studies of Shang et al. [25] and Erickson et al. [26].

mucus layer was assumed as one-dimensional (1-D) because the layer thickness is negligibly small compared to the scale of the whole nasal cavity. This allowed an analytical solution for the drug diffusion through mucus proposed by Erickson et al. [26] and is depicted in Fig 13 where a dissolved droplet is initially at the upper boundary of the mucus layer ($x = 0$) after deposition on the nasal cavity surface. Its motion is driven by Brownian diffusion, moving towards the lower absorbing boundary ($x = h_g$) due to a concentration gradient. The normalised concentration c(x, t) over time is governed by the classical diffusion equation [42],

$$\frac{\partial c}{\partial t} = D_g \frac{\partial^2 c}{\partial x^2}$$

(9)

where $D_g$ is the diffusion coefficient of the drug solute in mucus (gel layer) and t is the time after initial drug deposition. The boundary conditions were:

- reflective (zero-flux) boundary condition at the top of the gel layer, $\frac{\partial c}{\partial x}(0, t) = 0$;

- full-absorption boundary condition $c(h_g, t) = 0$ and;

- initial condition depicted by the Dirac's function $c(0, 0) = 2\delta(x)$,

  an infinite-series expression is obtained,

$$c(x, t) = \frac{2}{h_g} \sum_{n=0}^{+\infty} \left\{ e^{-\frac{(2n+1)^2 \pi^2}{4h_g^2} Dt} \times cos\left[\frac{(2n+1)\pi}{2h} x\right] \right\}$$

(10)

where $c(x, t)$ is the normalised concentration distribution over the gel layer depth at a time t after deposition. This is a classical solution to bounded diffusion problems using the standard separation of variables method [26, 42]. To evaluate the percentage of the drug particle absorbed by the tissue, an absorption function $A_s(t)$ was derived by integrating the probability

function from $x = 0$ to $x = h_g$,

$$A(t) = \left\{ 1 - \frac{4}{\pi} \sum_{n=0}^{+\infty} \left[ e^{-\frac{(2n+1)^2 \pi^2}{4h_g^2} Dt} \times \frac{(-1)^n}{2n+1} \right] \right\} \tag{11}$$

The diffusion coefficient of the solute in the mucus layer was estimated by the Obstruction-Scaling model [43],

$$\frac{D}{D_0} = e^{-\frac{\pi}{4} \left( \frac{r_s + r_f}{r_g + r_f} \right)^2} \tag{12}$$

where $r_f = 3.5nm$ is the mucin fibre radius, $r_g = 50nm$ is the mucin network's effective mesh fibre spacing [32], $r_s$ is the effective radius of the drug solute and $D_0$ is the corresponding solute diffusion coefficient in water, estimated by Stokes-Einstein equation:

$$D_0 = \frac{k_B T}{6\pi\mu_0 r_s} \tag{13}$$

$\mu_0$ is the water viscosity, $k_B$ is the Boltzmann's constant and T is the temperature.

## Conclusion

This study presents a CFD-based approach to evaluate nasal spray efficacy with mucociliary clearance. Particle locations visualized on the mapped nasal wall revealed that the majority of particles were cleared via the mainstream on the floor of the nasal cavity, which was indicated by the shared boundary between main passage and septum. The total drug absorption was peaked at 23.9% when drug solute radius reached $95nm$ and the drug absorption majorly occurred on the septum. As the solute radius became larger, the drug absorbed by the main passage and nasopharynx considerably decreased and increased, respectively. For solute radius smaller than $50nm$, the total drug absorption (nearly 20%) were mainly contributed by particles deposited on the septum (nearly 14%) and main passage (nearly 6%). For solute radius larger than $50nm$, however, the contributions from particles deposited on above two regions considerably dropped whereas the contribution from particles deposited on vestibule rapidly increased from nearly 0% to nearly 12%. This was because the mucus flow recirculation existed on the vestibule (no drug absorption) slowly moved particles out of vestibule (with drug absorption). As a longer time was required for larger solutes to diffuse through the mucus layer, it enabled more absorption to occur on the rest of the nasal cavity wall from particles originating from the vestibule.

## Author Contributions

**Conceptualization:** Kiao Inthavong, Fajiang He, Jiyuan Tu.

**Data curation:** Dasheng Qiu.

**Formal analysis:** Yidan Shang.

**Funding acquisition:** Yidan Shang, Narinder Singh, Fajiang He, Jiyuan Tu.

**Investigation:** Yidan Shang.

**Methodology:** Yidan Shang.

**Project administration:** Kiao Inthavong, Jiyuan Tu.

**Resources:** Dasheng Qiu, Narinder Singh, Fajiang He, Jiyuan Tu.

**Supervision:** Kiao Inthavong.

**Validation:** Yidan Shang.

**Visualization:** Yidan Shang.

**Writing – original draft:** Yidan Shang.

**Writing – review & editing:** Kiao Inthavong.

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
