## [Decision Letter · Decision Letter 0]

25 Nov 2020

PONE-D-20-28395

Prediction of nasal spray drug absorption influenced by mucociliary clearance

PLOS ONE

Dear Dr. Inthavong,

Thank you for submitting your manuscript to PLOS ONE. After careful consideration, we feel that it has merit but does not fully meet PLOS ONE’s publication criteria as it currently stands. Therefore, we invite you to submit a revised version of the manuscript that addresses the points raised during the review process.

We look forward to receiving your revised manuscript.

Kind regards,

Fang-Bao Tian

Academic Editor

PLOS ONE

Journal Requirements:

Reviewers' comments:

Reviewer's Responses to Questions

**Comments to the Author**

1. Is the manuscript technically sound, and do the data support the conclusions?

Reviewer #1: Yes

Reviewer #2: Yes

Reviewer #3: Yes

2. Has the statistical analysis been performed appropriately and rigorously? 

Reviewer #1: N/A

Reviewer #2: Yes

Reviewer #3: N/A

3. Have the authors made all data underlying the findings in their manuscript fully available?

Reviewer #1: Yes

Reviewer #2: Yes

Reviewer #3: Yes

4. Is the manuscript presented in an intelligible fashion and written in standard English?

Reviewer #1: Yes

Reviewer #2: Yes

Reviewer #3: Yes

5. Review Comments to the Author

Reviewer #1: Great paper on nasal spray drug absorption using computational modeling. Well written with insightful information that will advance knowledge of nasal spray absorption based on mucociliary clearance rate.

Reviewer #2: This study developed an advanced modelling framework for nasal spray drug absorption. The computational study consider the mucus diffusion which is essential for the more realistic drug absorption in nasal airways. Authors clearly outlined the research gaps in the introduction section. The research methodology is well defined. The novelty and significance of this study are enough to be accepted for publication in PlosOne. Authors may address the following issues in the revised manuscript;

1. Authors highlighted the research gaps in the introduction section. Authors may include a sentence at the end of the introduction section and write down how this study will advance the present understanding?

2. Please check the quality of figure 1.

3. Authors used the CT-Scan images and did not mention whether the ethical procedure is completed or not? This is a low-risk project and authors need to mention the ethical detail.

4. Authors may present some information regarding mesh size and quality for future reference.

5. Needs to check the quality of other figures.

6. Authors may explain the ‘trap’ condition for the reader.

7. Authors may mention some information regarding the viscous model.

8. Did authors consider cilia movement?

Reviewer #3: This work uses CFD-based approach to quantify regional drug deposition inside a nasal geometry and tracks the subsequent mucociliary transport of the deposited particles. The study then uses 1-D diffusion model to predict absorption of drug solute into the gel-like mucus upper layer. I have the following comments and questions for the authors:

1. This is a very interesting study and can provide a more realistic picture on therapeutic transmission to the epithelial cells, which can have wider clinical ramifications.

2. The authors should add more details on how the mucus source term was determined. Was it through a series of iterative simulations, so as to achieve the stated 6 mm/min averaged mucus velocity along the main cavity walls?

3. On page 10, the authors state that "the particles were assumed to immediately dissolve into macromolecules after depositing". I presume the "macromolecules" constitute the solute agents that are now on the mucus upper layer. At time t = 0, what was the area over which the solute agents were concentrated? And how does it relate to the size of the particle / droplet that delivered those solute agents?

4. What is the solute concentration in the drug particles / droplets? Is there a distribution for the 1-110 nm solute radii that are embedded / suspended in each droplet? These details seem to be missing in the manuscript.

5. I understand that an experimental validation of the solute transport trends might be out-of-scope for this manuscript. But it would be useful to the reader if the authors can elaborate on the possible validation approaches, as part of the discussion section.

6. Could you label Stream 1, Stream 2 etc. on Figure 4b? This is in context to the comments on page 4, second paragraph.

7. There are some typos in the manuscript. E.g. (a) in the abstract: ``The spray particle deposition distribution was validated experimentally and numerically, and the mucus velocity field''; (b) on page 14: ``The RSM The simulation was considered converged...''

In conclusion: I think the manuscript, as it is now, needs some minor revisions.

6. PLOS authors have the option to publish the peer review history of their article (what does this mean?). If published, this will include your full peer review and any attached files.

Reviewer #1: No

Reviewer #2: No

Reviewer #3: No

---

## [Author Response · Author response to Decision Letter 0]

9 Jan 2021

For detailed response, please see attached file "response to reviewers". Below is the text-only response.

Response to Reviewer #1's Comments

Reviewer #1: Great paper on nasal spray drug absorption using computational modeling. Well written with insightful information that will advance knowledge of nasal spray absorption based on mucociliary clearance rate.

Re: We appreciate positive comments from reviewer #1.

Response to Reviewer #2's Comments

Reviewer #2: This study developed an advanced modelling framework for nasal spray drug absorption. The computational study consider the mucus diffusion which is essential for the more realistic drug absorption in nasal airways. Authors clearly outlined the research gaps in the introduction section. The research methodology is well defined. The novelty and significance of this study are enough to be accepted for publication in PlosOne. Authors may address the following issues in the revised manuscript:

1. Authors highlighted the research gaps in the introduction section. Authors may include a sentence at the end of the introduction section and write down how this study will advance the present understanding?

Re: We appreciate the reviewer’s suggestions. Several sentences have been added to the end of the Introduction section to address how this study will advance the present understanding, as seen in page 2-3, line 38-42:

“This study advances current mucociliary clearance modelling by applying realistic spray droplets to determine preliminary deposition sites that allow more precise locations for evaluating drug absorption through nasal cavity wall. The mucus-diffusion modelling framework will provide a more realistic picture on therapeutic transmission to the epithelial cells.” 

2. Please check the quality of figure 1.

Re: The figures were uploaded through the PLOS One submission system and somehow resolutions of all figures were compressed. Originally, the Figure 1 was a high-quality TIF image with dimensions of 2250*1167 and resolution of 300 dpi. The original high-quality figures can be found in the “File Inventory” of the Editorial Manager System. In case they are not available for reviewers, the original Figure 1 is attached below and all high-quality figures are attached in the appendix section of the rebuttal.

3. Authors used the CT-Scan images and did not mention whether the ethical procedure is completed or not? This is a low-risk project and authors need to mention the ethical detail.

Re: The data has been approved by the ethics committee of the hospital to make sure all patients were well informed. All private information (e.g. name, address and ID) of patients were erased before uploading to our secured server. A sentence was added to the manuscript to clarify this in page 3, line 47-49:

“This research was approved by the institutional review board at the Hubei Cancer Hospital (Hubei Province, China), and informed consent was obtained from patients.” 

4. Authors may present some information regarding mesh size and quality for future reference.

Re: We agree that the information regarding mesh size and quality should be described for future reference. The relevant information was summarised in the “Mesh generation and boundary conditions” sub-section of the “Materials and methods” section, as seen below:

“The CFD domain of the nasal cavity was created with polyhedral mesh elements, refined near the boundaries, and in regions with high curvature and thin gaps, using Ansys-Fluent-Meshing v19.3 (Fig 12). Five prism layers were created on computational boundaries. One prism layer that was attached on the nasal cavity walls was extracted to form the computational domain of the mucus layer, which covered walls of the vestibule, the main cavity and the outlet extension pipe. A mesh independence test was performed for meshes under three refinements (e.g. coarse, medium and fine) and the optimal mesh (medium) with 2.02 million cells was selected for computation. The maximum skewness of the selected mesh was 0.70.”

5. Needs to check the quality of other figures.

Re: Similar with Question 2, the resolutions of all figures were compressed by the PLOS One submission system. Please check original figures in the file inventory of the Editorial Manager System or in the appendix section of the rebuttal.

6. Authors may explain the ‘trap’ condition for the reader.

Re: “Trap” is a classical boundary condition for DPM particle tracking. The description of the trap condition has been updated as seen in page 9, line 247-249:

“The DPM (Discrete Particle Method) condition at the nasal walls was set to “trap”, where the Lagrangian particle tracking was terminated the individual particle came into contact with the nasal walls.” 

7. Authors may mention some information regarding the viscous model.

Re: We agree that it is important to provide information about viscous models. In this study, two viscous models, Reynolds Stress Model (RSM) and Laminar models, were used for the airflow and mucus flow simulations, respectively. The laminar model does not involve turbulence modelling and hence no further information is needed. Regarding the RSM, complex equations involving turbulent diffusion, molecular diffusion, stress production, buoyancy production, pressure strain and dissipation. Listing all these equation components will be out-of-scope in this study. A sentence has been updated to expand the information of the viscous model, as seen in page 9, line 269-271:

“The Reynolds Stress Model (RSM), which has the advantage of capturing turbulent anisotropy near walls in a complex flow, was selected to predict the transitional flow using the SIMPLE algorithm for pressure-velocity coupling. The accuracy of RSM model has been evidenced to cover low and high Reynold's number ranges. In this study, the RSM simulation was considered converged when all residuals reached 1e-5.”

8. Did authors consider cilia movement?

Re: With assumptions stated in the “Mucus-diffusion model for dissolved particles” section, 

 “The high viscous gel eliminates convection in the upper gel layer, thus we ignored the velocity along the wall-normal direction, following Rygg and Longest [21]) and Shang et al. [25].”

 “Since the periciliary layer does not host the mucin network, the diffusion model was simplified with the assumption that the solute reaching the periciliary layer is immediately absorbed by the epithelium.”

this study bypassed simulating cilia movement. Therefore, cilia movement was not considered in this study.

 

Response to Reviewer #3's Comments

Reviewer #3: This work uses CFD-based approach to quantify regional drug deposition inside a nasal geometry and tracks the subsequent mucociliary transport of the deposited particles. The study then uses 1-D diffusion model to predict absorption of drug solute into the gel-like mucus upper layer. I have the following comments and questions for the authors. In conclusion: I think the manuscript, as it is now, needs some minor revisions.

1. This is a very interesting study and can provide a more realistic picture on therapeutic transmission to the epithelial cells, which can have wider clinical ramifications.

Re: We appreciate the reviewer’s suggestions. Following sentence has been added to the manuscript in page 2-3, line 38-42:

“This study advances current mucociliary clearance modelling by applying realistic spray droplets to determine preliminary deposition sites that allow more precise locations for evaluating drug absorption through nasal cavity wall. The mucus-diffusion modelling framework will provide a more realistic picture on therapeutic transmission to the epithelial cells.” 

2. The authors should add more details on how the mucus source term was determined. Was it through a series of iterative simulations, so as to achieve the stated 6 mm/min averaged mucus velocity along the main cavity walls?

Re: We agree with the reviewer’s suggestions. To clarify the how the mucus flow was simulated, we added following statements to the manuscript in page 9, line 262-265:

“The value S_mucus was determined by allowing it to produce an average mucus velocity magnitude of 6 mm/min[35] on the main cavity wall that excluded vestibules and sinuses…The accumulation of the mucus source term and the incompressible assumption pushed the mucus backwards to the nasopharynx region…The low mucus velocity led to a Reynold's number as low as 1e-4, hence the mucus flow was considered as laminar flow. The mucus velocity distribution was calculated iteratively using second-order discretisation method and the SIMPLE scheme.” 

3. On page 10, the authors state that "the particles were assumed to immediately dissolve into macromolecules after depositing". I presume the "macromolecules" constitute the solute agents that are now on the mucus upper layer. At time t = 0, what was the area over which the solute agents were concentrated? And how does it relate to the size of the particle / droplet that delivered those solute agents?

Re: The reviewer was correct that after depositing, the macromolecules constitute the solute agents were located at the mucus upper layer, at time t=0. In terms of the “where the solute agents were concentrated”, the initial deposition patterns of spray particles were visualised in Figure 5 and the descriptions were in page 4-5, line 89-103:

“…There was no deposition in the maxillary sinuses, and therefore these regions were removed, but the ostia openings were retained. There are three deposition hot-spots located at the posterior vestibule (�), anterior septum (�) and the anterior tip of the middle turbinate (�) for large particles with sizes > 60µm, whereas the anterior main nasal passage (�) are the main deposition locations for relatively smaller particles with sizes nearly 20µm. There is another minor hot-spot observed near the maxillary ostia (�) for nearly 30µm particles.

The deposition efficiency on the vestibule wall and the main passage walls were compared with in-vitro measurements[31] and CFD results[21](Fig 6.). Our results produced 100% deposition efficiency in the nasal cavity, with approximately 79.8%, 19% and 1.2% particle mass deposition in the vestibule region, main passages and nasopharynx, respectively. The deposition efficiency in the vestibule region is 2.3% higher than the results from Rygg and Longest[21] and 14% higher than from Azimi et al.[31].” 

4. What is the solute concentration in the drug particles / droplets? Is there a distribution for the 1-110 nm solute radii that are embedded / suspended in each droplet? These details seem to be missing in the manuscript.

Re: We appreciate the reviewer’s suggestions. We agree that the solute concentration in the drug droplets is critical for clinical applications. However, this is out-of-scope in this study. Our mucus-drug evaluation modelling only evaluate the fraction of the drug solute being absorbed. In order to link the evaluation model with clinical use, new statements and a new equation have been added to the Discussion section, as seen in page 7-8, line 214-218:

“There exists a gap between the absorption fraction and the real dosage. Equation (3) is required to link the drug absorption fraction with drug dosage in the clinical applications.

dosage=4/3 π×c×∑_(i=1)^n▒〖[A_i (+∞)×r_i^3]〗 (3)

where c is the solute concentration, A_i (+∞) is the final absorption fraction of the ith deposited spray droplet and r^i is its radius.”

5. I understand that an experimental validation of the solute transport trends might be out-of-scope for this manuscript. But it would be useful to the reader if the authors can elaborate on the possible validation approaches, as part of the discussion section.

Re: We agree with the reviewer’s suggestions about validations. In this study, validations have been performed from several aspects. Firstly, the spray droplet size distribution depicted by the Rosin-Rammler distribution was validated by measured data from Dayal et al. (2004). Secondly, spray droplet deposition distributions on nasal cavity wall were validated by measured data from Azimi et al. (2015). Lastly, the method to create the mucus velocity distribution has been validated in the previous study Shang et al. (2019). To clarify this, a new sentence has been added to the Discussion section in page 7, line 191-192:

“…The mucus velocity field simulated in this study is consistent with the 3D-shell model in the previous work [25] except for a slightly stronger radial mucus flow observed around ostium. Therefore the mucus velocity distribution can be considered validated. In this study, the vestibule…” 

6. Could you label Stream 1, Stream 2 etc. on Figure 4b? This is in context to the comments on page 4, second paragraph.

Re: We appreciate the reviewer’s comments. We updated an old image without texts of stream numbers by mistake. The Figure 4 has been corrected as shown below:

7. There are some typos in the manuscript. E.g. (a) in the abstract: ``The spray particle deposition distribution was validated experimentally and numerically, and the mucus velocity field''; (b) on page 14: ``The RSM The simulation was considered converged...''

Re: We appreciate the reviewer’s suggestions. Two sentences have been updated as following:

(1). “The spray particle deposition distribution was validated experimentally and numerically, and the mucus velocity field was validated by comparing with previous studies.”

(2). “The RSM simulation was considered converged when all residuals reached 1e-5.”

---

## [Editor Report · Decision Letter 1]

12 Jan 2021

Prediction of nasal spray drug absorption influenced by mucociliary clearance

PONE-D-20-28395R1

Dear Dr. Inthavong,

We’re pleased to inform you that your manuscript has been judged scientifically suitable for publication and will be formally accepted for publication once it meets all outstanding technical requirements.

Kind regards,

Fang-Bao Tian

Academic Editor

PLOS ONE

Additional Editor Comments (optional):

Thank you for revising your work, which now is acceptable for publication in the Plos One.
---

## [Editor Report · Acceptance letter]

18 Jan 2021

PONE-D-20-28395R1 

Prediction of nasal spray drug absorption influenced by mucociliary clearance 

Dear Dr. Inthavong:

I'm pleased to inform you that your manuscript has been deemed suitable for publication in PLOS ONE. Congratulations! Your manuscript is now with our production department. 

Kind regards, 

on behalf of

Dr. Fang-Bao Tian 

Academic Editor

PLOS ONE